# Time-Dependent Effect of Eggshell Membrane on Monosodium-Iodoacetate-Induced Osteoarthritis: Early-Stage Inflammation Control and Late-Stage Cartilage Protection

**DOI:** 10.3390/nu16121885

**Published:** 2024-06-14

**Authors:** Min Yu, Cheoljin Park, Young Bae Son, So Eun Jo, Seong Hee Jeon, Ye Jin Kim, Sang Bae Han, Jin Tae Hong, Dong Ju Son

**Affiliations:** College of Pharmacy, Chungbuk National University, 194-21 Osongsaengmyong 1-ro, Osong-eup, Heungduk-gu, Cheongju 28160, Chungbuk, Republic of Korea; yumin0413@chungbuk.ac.kr (M.Y.); cusimquar@hanmail.net (C.P.); yb@chungbuk.ac.kr (Y.B.S.); lovel5508@chungbuk.ac.kr (S.E.J.); shee@chungbuk.ac.kr (S.H.J.); yejinee13@chungbuk.ac.kr (Y.J.K.); shan@chungbuk.ac.kr (S.B.H.); jinthong@chungbuk.ac.kr (J.T.H.)

**Keywords:** osteoarthritis, eggshell membrane, Ovomet, inflammation, cartilage-degrading enzyme

## Abstract

Osteoarthritis (OA) is a chronic degenerative joint disease that causes chronic pain, swelling, stiffness, disability, and significantly reduces the quality of life. Typically, OA is treated using painkillers and non-steroidal anti-inflammatory drugs (NSAIDs). While current pharmacologic treatments are common, their potential side effects have prompted exploration into functional dietary supplements. Recently, eggshell membrane (ESM) has emerged as a potential functional ingredient for joint and connective tissue disorders due to its clinical efficacy in relieving joint pain and stiffness. Despite promising clinical evidence, the effects of ESM on OA progression and its mechanism of action remain poorly understood. This study evaluated the efficacy of Ovomet^®^, a powdered natural ESM, against joint pain and disease progression in a monosodium iodoacetate (MIA)-induced rodent model of OA in mice and rats. The results demonstrate that ESM significantly alleviates joint pain and attenuates articular cartilage destruction in both mice and rats that received oral supplementation for 5 days prior to OA induction and for 28 days thereafter. Interestingly, ESM significantly inhibited mRNA expression levels of pro-inflammatory cytokines including tumor necrosis factor alpha (TNF-α), interleukin-1β (IL-1β), and interleukin-6 (IL-6), as well as inflammatory mediators, cyclooxygenase-2 (COX-2), and inducible nitric oxide synthase in the knee joint cartilage at the early stage of OA, within 7 days after OA induction. However, this effect was not observed in the late stage at 28 days after OA induction. ESM further attenuates the induction of protein expression for cartilage-degrading enzymes like matrix metalloproteinase (MMPs) 3 and 13, and a disintegrin and metalloproteinase with thrombospondin motifs 5 (ADAMTS-5), in the late-stage. In addition, MIA-induced reduction of the protein expression levels of cartilage components, cartilage oligomeric matrix protein (COMP), aggrecan (ACAN) and collagen type II α-1 chain (COL2α1), and cartilage extracellular matrix (ECM) synthesis promoting transcriptional factor SRY-Box 9 (SOX-9) were increased via ESM treatment in the cartilage tissue. Our findings suggest that Ovomet^®^, a natural ESM powder, is a promising dietary functional ingredient that can alleviate pain, inflammatory response, and cartilage degradation associated with the progression of OA.

## 1. Introduction

Osteoarthritis (OA), a degenerative joint disease most prevalent in older people, arises from articular cartilage destruction and inflammation [1,2]. It affects various joints, especially the knee, leading to joint pain, decreased mobility, and a decline in quality of life for patients [3,4,5]. Cartilage breakdown, a hallmark of OA, results from chondrocyte dysfunction [6]. The articular cartilage extracellular matrix (ECM) is composed primarily of collagen type II and aggrecan, a major proteoglycan, which provide strength and elasticity [7,8,9,10,11]. The degradation of this ECM depends on the balance between anabolic and catabolic activities [12]. Pro-inflammatory factors released by the inflamed synovium trigger chondrocytes to degrade the cartilage ECM, ultimately destroying it [12,13]. Therefore, restoring the balance between anabolism and catabolism is a critical therapeutic target in OA.

Non-steroidal anti-inflammatory drugs (NSAIDs) are commonly used for their ability to rapidly reduce pain symptoms in OA. However, long-term use of NSAIDs can lead to various adverse effects on the gastrointestinal, kidney, and cardiovascular systems [14,15]. Despite these risks, NSAIDs remain widely used due to the lack of effective and safe alternatives. This highlights the need for safer OA management strategies, leading researchers to explore the efficacy of natural products [16]. Several functional ingredients have been widely studied and used for promoting joint health and managing OA, particularly those proposed to protect articular cartilage. These include glucosamine, methylsulfonylmethane (MSM), chondroitin sulfate, collagen hydrolysates, pycnogenol, apigenin, resveratrol, curcumin, avocado–soybean unsaponifiables, willow bark extract, *Boswellia serrata* extract, and *Arnica montana* extract [16,17].

Eggshell membrane (ESM) is a unique, protein-rich, double-layered fibrous meshwork located between the eggshell and the egg white. It is composed of approximately 500 different types of proteins [18,19]. The major component of ESM is collagen, with types I, V, and X constituting about 10% of it dry weight [20]. Other key components include fibronectin, osteopontin, chondroitin sulfate, and hyaluronic acid (HA) [21]. Despite its unique properties and composition of many beneficial substances, ESM has been underutilized in the food industry compared to other egg components, such as egg white and yolk [21]. Recent studies, however, have reported beneficial effects of ESM, including antioxidant and anti-inflammatory activities, the promotion of gut health, and skin wound healing [22,23,24,25,26,27,28,29,30]. Beyond its antioxidant and other health benefits, ESM shows particular promise in improving joint health. Studies have shown ESM to relieve pain associated with OA [31,32,33,34,35]. Ruff et al. reported that ESM significantly reduced joint pain and stiffness in postmenopausal female [31]. Kiers et al. also found that daily consumption of ESM by OA patients led to considerable relief in keen pain [32]. Additionally, ESM has been shown to be effective in improving joint function and pain relief in elderly PA patients and CrossFit athletes [33,34,35]. A recent clinical trial using a water-soluble ESM hydrolysate supplement showed improvement in knee joint function and mobility among OA patients, particularly those with severe conditions [36]. Another study explored the potential use of ESM in tissue engineering of cartilage substitutes. The combination of ESM with silk fibroin and polyvinyl alcohol exhibited properties similar to human meniscus cartilage and promoted cell growth of articular chondrocytes [37].

Although previous studies have shown beneficial properties for ESM in reducing OA symptoms and improving joint health, the exact mechanisms for ESM in disease progression are not yet fully understood. This study, therefore, investigated the efficacy of a powdered natural ESM supplement, Ovomet^®^, against joint pain and disease progression in a monosodium iodoacetate (MIA)-induced rodent model of OA in mice and rats.

## 2. Materials and Methods

### 2.1. Preparation of ESM

A powdered natural ESM (Ovomet^®^, Lot number 070251AS) was produced and provided by Eggnovo S.L. (Navarra, Spain). In brief, the eggshells were washed with water and the ESM was carefully separated. The separated ESM was dried with hot air and sterilized. Subsequently, it was milled to a micronized size under 30 μm. This powdered ESM material contains 8 mg/g of hydroxyproline.

### 2.2. Animals and Ethics Statement

Six-week-old male C57BL/6J mice (body weight: 21–23 g) and Sprague Dawley (SD) rats (body weight: 160–180 g) were obtained from Daehan Bio Link Co., Ltd. (Eumseong, Chungbuk, Republic of Korea) and housed in the animal facility at the College of Pharmacy, Chungbuk National University (Cheongju, Chungbuk, Republic of Korea). Animals were acclimated for 7 days and maintained under conventional housing conditions at 23 ± 2 °C with a controlled 12 h light–dark cycle and were provided filtered tap water and a rodent chow diet (Rodfeed^®^, Daehan Bio Link) ad libitum throughout the experiment.

All animal experimental procedures complied with the *National Institutes of Health Guide for the Care and Use of Laboratory Animals* and the *Korean National Animal Welfare Law*. The Institutional Animal Care and Use Committee of Chungbuk National University reviewed and approved all animal experimental protocols (IACUC approval number CBNUA-1564-21-02).

### 2.3. Experimental Design and Administration

In this study, we used two species of rodent to determine potential differences in the effect of treatment on different biological systems and to better reflect the intended application in humans: mice for pain assessment, histology, and mRNA expression and rats for pain assessment, gross pathology, protein expression, and cartilage-degrading marker analysis.

The design of the experiment for each animal species, mouse or rat, is detailed in Table 1 and Figure 1. Each of the five groups (*n* = 30 animals per group) consisted of a normal Sham control (Sham), an MIA-induced OA control (MIA), a positive control (CLX), and two groups receiving increasing doses of ESM (total: 150 mice and 150 rats). The ESM doses for each animal species were determined by converting the recommended human equivalent daily doses of 300 and 600 mg following a practice guide for dose conversion between animals and human [38]. Experimental groups and doses of ESM and celecoxib (CLX) are presented in Table 1. All groups received daily oral gavage treatment for five days before OA induction and continued 28 days post-induction until the experiment’s end. The Sham and MIA groups received equivalent volumes of vehicle (normal saline). Animals were closely monitored for general health conditions and other clinical complications. Joint pain was assessed using weight-bearing measurements of hind limbs on designated days, and animals were euthanized via CO_2_ inhalation at the designated endpoints for tissue or blood sample collection (Figure 1).

### 2.4. MIA-Induced OA Animal Modelling

MIA-induced OA animal modelling was conducted as described previously [39,40]. In brief, animals were anesthetized via inhalation of 2.0–3.5% isoflurane (Piramal Critical Care, Bethlehem, PA, USA). Then, the animals received an intra-articular injection of MIA at a dose of 1 mg/10 μL for mice and 3 mg/50 μL for rats, respectively, on their left hind limb gap right below the patella to induce OA. The Sham group received the same volume of normal saline. Following injection, the animals were allowed to fully recover from anesthesia and were monitored appropriately before returning them to their cages. They then continued to receive the same oral doses of ESM, CXL, or vehicle, respectively, until their designated endpoints.

### 2.5. Pain Assessment (Weight-Bearing Test)

Joint pain was assessed by measuring weight-bearing distributions between the postoperative and normal hind limbs using an incapacitance meter (600MR, IITC Life Science, Inc., Woodland Hills, CA, USA), as described previously [41,42]. Animals were placed in an angled acryl chamber facing forward. The left hind paw (ipsilateral—on the same side as the OA-induction) and the right hind paws (contralateral—on the opposite side) were each placed on a separate force sensor in the center. Each measurement lasted for 5 s, and the weight-bearing value was obtained by averaging three repeated measurements. These values were converted into a weight distribution percentage (%) by dividing the ipsilateral weight by the sum of the ipsilateral and contralateral weights, according to the calculation formula below.
$$\mathrm{Weight}\;\mathrm{bearing}\;\left(\%\right)=\frac{\left(\mathrm{Weight}\;\mathrm{bearing}\;\mathrm{of}\;\mathrm{left}\;\mathrm{hind}\;\mathrm{limb}\;\mathrm{average}\right)}{\mathrm{Weight}\;\mathrm{bering}\;\mathrm{of}\;\mathrm{left}\;\mathrm{hindlimb}\;\mathrm{average}\;+\;\mathrm{Weight}\;\mathrm{beraing}\;\mathrm{of}\;\mathrm{right}\;\mathrm{hindlimb}\;\mathrm{everage})}\times 100\;$$

### 2.6. Macroscopic Observation (Gross Pathology)

On days 14 and 28 post-OA induction, the left hind limbs of rats underwent microdissection to isolate joint tissues. Tendons and surrounding tissues were carefully removed from the specimen to expose the femoral cartilages for macroscopic observation. Cartilage surface images were then captured using a stereo-microscope (SZX7, Olympus LS, Tokyo, Japan) equipped with a digital camera.

### 2.7. Histological Analysis

On 28 days post-OA induction, the left hind limbs of mice were dissected, fixed in 4% paraformaldehyde solution (Biosesang, Seongnam, Republic of Korea) for 24 h at 4 °C, decalcified with 5% formic acid for 2 weeks, and then dehydrated in graded acetone and embedded in paraffin. Paraffin-embedded specimens were sectioned at 4–5 μm thickness using a microtome (CM 1850, Leica Microsystems, Wetzlar, Germany). Sections were stained with hematoxylin and eosin (H&E) solution (Leica Microsystems), a Safranin-O/Fast Green staining kit (IHC World, Woodstock, MD, USA), and a Toluidine Blue staining kit (IHC World) to evaluate structural cartilage damage. After mounting with Permount mounting medium (Fisher Scientific International, Inc., Pittsburg, PA, USA), stained section images were captured under a light microscope (Microscope Axio Imager. A2, Carl Zeiss, Oberkchen, Germany) at 5**×** and 20**×** magnification. All stained slides were then evaluated for cartilage damage using a modified Mankin scoring system (0–13 scale) via double-blind observation [43].

### 2.8. Immunohistochemistry

Paraffin-embedded sections from mice joint tissue were deparaffinized and rehydrated. Immunohistochemistry was performed to detect cartilage-degrading markers, cartilage oligomeric matrix protein (COMP), and aggrecan (ACAN). Antigen retrieval was achieved via incubation in a citrate buffer (10 mM sodium citrate, pH 6.0) at in 60 °C for 20 min. Sections were then rinsed with phosphate-buffered saline (PBS) and blocked with 10% donkey serum in PBS. Immunohistochemical staining was carried out using primary antibodies: anti-COMP (Cat No. ab231977; 1:100, abcam, Waltham, MA, USA) or anti-ACAN (Cat No. 138801-1AP; 1:150, Proteintech, Rosemont, IL, USA) incubated overnight at 4 °C. Following primary antibody incubation, sections were washed and incubated with biotinylated secondary antibodies for 2 h at room temperature. Immunocomplex visualization was performed using the DAB Detection IHC kit (abcam) according to the manufacturer’s instructions. Stained sections were imaged under a light microscope (Microscope Axio Imager. A2; Carl Zeiss) at 5× and 40× magnification.

### 2.9. Real-Time PCR Analysis

Total RNA was isolated from mouse articular cartilage tissues on days 1, 7, and 28 after OA induction using the Hybrid-R Total RNA isolation kit (GeneAll Biotechnology, Seoul, Republic of Korea) according to the manufacturer’s protocol. The isolated RNA was then reverse transcribed into complementary DNA (cDNA) using the High-Capacity cDNA Reverse Transcription kit (Applied Biosystems, Foster City, CA, USA). Quantitative real-time PCR (RT-qPCR) was performed using the THUNDERBIRD™ SYBR^®^ qPCR mater mix, (Toyobo, Osaka, Japan) with custom-designed primers, using β-actin as the housekeeping control on QuantStudio 3 RT-qPCR system (Applied Biosystems). The primer sequences are listed in Table 2. Relative fold changes in gene expressions were determined using the 2**^−^**^ΔΔCT^ method.

### 2.10. Western Blot Analysis

Total protein was extracted from rat articular cartilage tissue (tibial plateau) dissected on 28 post-OA induction. Approximately 40 mg of tissue was homogenized in 500 μL of lysis buffer containing 50 mM Tris-HCl (pH 7.4), 150 mM NaCl, 2 mM EDTA, 1% NP-40, 0.5% deoxycholate, and 0.1% SDS using the Bead Ruptor Elite (Omni International, Kennesaw, GA, USA). Following homogenization, the lysates were incubated overnight at 4 °C. Supernatants were collected via centrifugation at 4 °C, 12,500 rpm for 20 min. Total protein concentration in the lysates was measured using the Micro BCA™ Protein Assay kit (Thermo Scientific, Waltham, MA, USA). Thirty micrograms of protein were loaded and separated on 8% or 10% Tris-glycine polyacrylamide gel. The separated proteins were then transferred to polyvinylidene fluoride (PVDF) membranes. After blocking with 5% skim milk, the membranes were probed with primary antibodies against: matrix metalloproteinase (MMP)-3 (Cat No. sc-21732; 1:1000, Santa Cruz biotechnology, Dallas, TX, USA), MMP-13 (Cat No. 8165-1-AP; 1:1000, Proteintech), a disintegrin and metalloproteinase with thrombospondin motifs 5 (ADAMTS-5) (Cat No. ab41037; 1:1000, abcam), collagen type II α-1 chain (COL2A1) (Cat No. sc-52658; 1:1000, Santa Cruz biotechnology), SRY-Box transcription factor 9 (SOX-9) (Cat No. ab185966; 1:1000, abcam), and β-actin (Cat No. 47778; 1:1000, Santa Cruz biotechnology). Following primary antibody incubation, the membranes were washed and incubated with HRP-conjugated secondary antibodies. Protein bands were visualized using enhanced chemiluminescence (ECL) detection with Clarity Max Western ECL substrate (Bio-Rad, Hercules, CA, USA) and imaged using the ChemiDoc™ Imaging system (Bio-Rad). Band intensities were quantified using Image Lab 4.1 Software (Bio-Rad).

### 2.11. Glycosaminoglycan (GAG) Assay

Quantitative assessment of GAG content in rat femoral groove tissues was performed using the Blyscan™ GAG assay kit (Cat No. B1000, Biocolor, Carrickfergus, UK) according to the manufacturer’s instruction. In brief, tissues were dissected from the joint on days 3, 7, and 28 post-OA induction, cut into pieces weighing approximately 25–35 mg, and stored at –80 °C until analysis. Samples were then digested in papain (125 µg/mL, Sigma-Aldrich, St. Louis, MO, USA) in 50 mM phosphate buffer (pH 6.5) containing 2 mM N-acetyl cysteine, for 18 h at 60 °C. Following digestion, the sulfated GAG content was measured using the assay kit. Total GAG concentration was quantified using a microplate reader at 656 nm.

### 2.12. Enzyme-Linked Immunosorbent Assay (ELISA)

To assess cartilage degradation, serum levels of COMP and the C-terminal cross-linked telopeptide of type II collagen (CTX-II) were measured in rats using ELISA on days 3, 7, and 28 post-OA induction. Blood samples were collected and centrifuged at 4 °C, 1500× *g* for 15 min to separate the serum. The isolated serum aliquots were then stored at –80 °C until further analysis. Commercially available ELISA kits were used to quantify COMP (Cat No. abx256440, Abbexa, Cambridge, UK) and CTX-II (Cat No. EEL-R2554, Elabscience Biotechnology, Houston, TX, USA) according to the manufacturer’s instructions.

### 2.13. Statistical Analysis

All data are presented as mean ± standard error of measurement (SEM) with the number of experiments indicated for each data point. The statistical significance of difference between groups was determined using either unpaired Student’s *t*-test for comparisons between two groups or repeated-measure (RM), two-way ANOVA followed by Dunnett’s post hoc test for groups across time points. All statistical analyses were performed using GraphPad Prism software version 8.4.2 (GraphPad Software, Boston, MA, USA).

## 3. Results

### 3.1. ESM Exerted Pain-Alleviating Effects in MIA-Induced OA

We first investigated the effect of oral ESM administration on joint pain in MIA-induced OA rodent models. OA is characterized by pain, and the severity of pain can be assessed by measuring the weight distribution between the ipsilateral (affected) and contralateral (unaffected) hind limbs. In both the mouse (Figure 2A) and rat models (Figure 2B), the MIA group exhibited significantly lower weight-bearing distribution compared to the Sham group throughout the experimental period, indicating increased pain. Conversely, animals in the ESML-, ESMH-, and CLX-treated groups displayed greater weight distribution on the affected paw (increased weight-bearing) compared to the MIA group. Notably, the high-dose of ESM groups in both mice and rats showed significantly higher weight distribution compared to the MIA group. These findings demonstrate that oral administration of ESM has the potential to alleviate joint pain in MIA-induced OA animals.

### 3.2. ESM Protects Articular Cartilage in MIA-Induced OA

To investigate the effect of oral ESM administration on cartilage destruction in MIA-induced OA, a macroscopic evaluation of the femoral condyles was performed on days 14 and 28 post-OA induction in a rat model. The Sham group exhibited a smooth articular surface, as shown in Figure 3 (first column on the left). In contrast, the MIA group displayed severe cartilage damage on days 14 and 28, characterized by large, irregular osteophytes; sclerosis; and subchondral bone exposure due to chondral erosions (Figure 3, second column from the left). Compared to the MIA group, treatment with CLX resulted in a reduction in osteophyte formation at both time points (Figure 3, third column from the left). Similarly, both the low-dose (ESML) (Figure 3, fourth column from the left) and high-dose (ESMH) (Figure 3, fifth column from the left) ESM groups displayed significantly reduced osteophyte formation. Additionally, the CLX, ESML, and ESMH groups all exhibited less severe subchondral bone exposure compared to the MIA group, indicating a protective effect against chondral erosion.

To further investigate the protective effects of ESM on articular cartilage, histological analysis was performed on a mouse model of MIA-induced OA. Three staining techniques were employed: H&E, Safranin O/Fast Green, and Toluidine Blue. H&E staining assessed the overall cartilage structure. The Sham group exhibited a smooth cartilage surface with distinct superficial, middle, and deep zones. Chondrocytes appeared in liner columns within lacunae. In contrast, the MIA group displayed irregular cartilage with fibrosis and increased abnormal chondrocytes. Notably, the CLX and ESM groups showed a marked reduction in these degenerative changes (Figure 4A, upper panels). Safranin O/Fast Green (Figure 4A, middle panels) and Toluidine Blue (Figure 4A, lower panels) staining evaluated proteoglycan content, a key structural component of cartilage. In healthy cartilage, these stains would intensely stain the entire region, from the superficial to the deep zone. However, due to cartilage loss in the MIA groups, Safranin O/Fast Green and Toluidine Blue staining intensity was weak. In contrast, compared to the MIA group, the CLX and ESM groups exhibited significantly stronger staining, suggesting a protective effect against proteoglycan loss. To quantify these observations, OA severity was graded using the modified Mankin scoring system by trained observers (Figure 4B). Scores for structure, cells, and staining were all lower in the CLX and all ESM groups compared to the MIA group. Notably, the total Mankin score was significantly lower in the high-dose ESM group (ESMH) compared to the MIA group. These findings indicate that oral administration of ESM can protect against cartilage loss in a mouse model of MIA-induced OA.

Taken together, our findings in both mouse and rat models indicated that oral administration of ESM can protect against cartilage destruction in MIA-induced OA.

### 3.3. ESM Suppresses Early Inflammatory Responses in MIA-Induced OA

Inflammation is a hallmark of OA and is thought to contribute to disease progression. Therefore, we investigated the effect of oral ESM administration on the expression of pro-inflammatory cytokines (TNF-α, IL-1β and IL-6) and inflammatory mediators (COX-2 and iNOS), depending on the stage of disease progression, in a mouse model of MIA-induced OA. As shown in Figure 5 and Appendix A, mRNA expression levels of IL-1β, IL-6, COX-2, and iNOS exhibited dose-dependent reductions compared to the MIA group on day 1 (early stage) post-OA induction. Notably, the ESMH group showed significantly lower expression of IL-6, COX-2, and iNOS mRNA compared to the MIA group (Figure 5). By day 7, mRNA expression of all pro-inflammatory cytokines (TNF-α, IL-1β, and IL-6) and COX-2 was significantly reduced in the ESMH group compared to the MIA group. However, by day 28 (late stage), no significant differences in gene expression were observed among the groups. These findings suggest that oral administration of ESM effectively reduces the inflammatory response in the early stages of MIA-induced OA.

### 3.4. ESM Inhibits the Expression of Cartilage-Degrading Enzymes in the Late Stage of MIA-Induced OA

MMPs and ADAMTS are key enzymes responsible for cartilage degradation during OA progression. To investigate the effect of ESM on these enzymes, we analyzed protein expression in the cartilage of MIA-induced OA rats on day 28 (late stage). Western blot analysis, as shown in Figure 6 and further detailed in Appendix A, revealed increased protein levels of MMP-3, MMP-13, and ADAMTS-5 in the MIA group compared to the Sham group. Notably, the ESML and ESMH groups displayed a trend of decreased MMP-3 and MMP-13 protein expression compared to the MIA group. Importantly, ADAMTS-5 protein expression showed a dose-dependent decrease in the ESM groups compared to the MIA group. These findings suggest that oral administration of ESM can effectively inhibit cartilage degradation by downregulating the expression of cartilage-degrading enzymes in the late stage of MIA-induced OA.

### 3.5. ESM Promotes Cartilage Repair in the Late Stage of MIA-Induced OA

We investigated the effect of oral ESM administration on cartilage health in MIA-induced OA mice and rats. Immunohistochemistry analysis of cartilage samples from mice on day 28 (late stage) revealed contrasting patterns of COMP, a marker of healthy cartilage (Figure 7A). The Sham group displayed strong COMP staining in the superficial and deep zones, whereas the MIA group exhibited irregular staining indicative of cartilage degradation. Notably, the COMP staining pattern in the ESM-treated groups (ESML and ESMH) resembled that of the Sham group, suggesting a protective effect of ESM.

Western blot analysis of cartilage from rats on day 28 further supports these findings (Figure 7B). MIA induction did not significantly alter COL2α1 protein levels compared to the Sham group. However, ESM treatment in MIA-induced OA rats resulted in a marked increase in COL2α1 expression, suggesting its potential to stimulate cartilage matrix production. Additionally, SOX-9, a key transcription factor for cartilage development, showed reduced protein expression in the MIA group compared to Sham. Notably, the high-dose ESM group (ESMH) exhibited increased SOX-9 protein expression, indicating potential promotion of cartilage repair via ESM.

We also analyzed glycosaminoglycan (GAG) content, a marker of cartilage health, in rat cartilage. Additionally, we examined serum levels of COMP and CTX-II, cartilage degradation biomarkers, in rats. GAG content in MIA group rat cartilage was significantly lower on days 3 and 7 compared to the Sham group. The ESM group showed a slight increase in GAG content on these days. On day 28 (late stage), GAG content was slightly elevated in the MIA and low-dose ESM (ESML) groups compared to Sham, while the high-dose ESM group (ESMH) showed a decrease compared to MIA (Appendix A).

Serum levels of COMP and CTX-II exhibited a dynamic response. Both markers were lower in the MIA group on day 3 compared to Sham but increased on days 7 and 28. ESM administration appeared to modulate these levels. The serum levels of COMP and CTX-II in the ESM groups were similar to the Sham group on day 3, suggesting a protective effect. On day 7, ESM groups displayed a dose-dependent increase in serum COMP and CTX-II levels, but not on day 28 (Appendix A).

Taken together, our findings in both mouse and rat models indicate that oral administration of ESM promotes cartilage repair in MIA-induced OA.

## 4. Discussion

OA is a debilitating disease characterized by pain and joint inflammation. This study investigated the effect of oral ESM administration on these crucial aspects of OA in MIA-induced mice and rats. Our results demonstrated that ESM treatment, particularly at high doses (ESMH), significantly alleviated pain in both animal models. This was evidenced by increased weight-bearing distribution on the affected paw compared to the MIA control group. These findings support the potential of ESM as an analgesic for OA patients.

Further strengthening this notion, ESM administration in mice specifically targeted the early stage of OA (day 1) by downregulating the expression of pro-inflammatory cytokines (TNF-α, IL-1β, IL-6) and inflammatory mediators (COX-2, iNOS) in the cartilage. These pro-inflammatory molecules are known to play a critical role in pain sensitization and joint destruction in OA [44,45]. Our data suggests that ESM’s ability to suppress these early inflammatory mediators may contribute to its pain-relieving effects. This aligns with previous studies demonstrating the anti-inflammatory properties of ESM in human immune cells, with mechanisms potentially involving downregulation of the Toll-like receptor (TLR)-4 and NF-κB signaling pathways [23]. Notably, a study by Vuong et al. also reported that ESM decreased the levels of pro-inflammatory cytokines IL-1β and IL-6 in human immune cells, suggesting a potential shared mechanism for in vivo and in vitro observations [23].

Beyond pain relief, this study explored the potential of ESM to protect cartilage from degeneration in MIA-induced OA. We employed various methods, including histological analysis (H&E, Safranin O/Fast Green, Toluidine Blue) with Mankin scoring and protein expression studies, to assess the impact of ESM treatment on cartilage health. Our findings revealed that ESM administration significantly inhibited the development of characteristic OA pathologies such as osteophyte formation and subchondral bone exposure. Additionally, ESM treatment reduced the loss of proteoglycans, a crucial component of healthy cartilage that contributes to its resilience and shock-absorbing properties. These observations suggest that ESM may help preserve the structural integrity of cartilage in OA joints.

Mechanistically, we investigated the effect of ESM on cartilage degradation and anabolism. ESM treatment downregulated the expression of matrix metalloproteinases (MMP-3, 13) in the late stage of OA (day 28) in rats. MMPs are enzymes known to break down cartilage components, and their excessive activity is a hallmark of OA progression [46]. By inhibiting MMP expression, ESM may help slow down cartilage degradation. Our data align with previous studies demonstrating that ESM from chicken eggs (similar to the ESM used here) can decrease the serum levels of MMP-2 and 9 in MIA-induced OA rats [47]. Furthermore, ESM treatment increased the expression of cartilage components (COL2α1, ACAN) and transcription factor SOX-9. COL2α1 is a major protein constituent of type II collagen, the primary structural component of cartilage. ACAN is a large proteoglycan molecule that helps attract water into cartilage, providing lubrication and resilience. SOX-9 plays a critical role in stimulating the synthesis of these essential cartilage components [13]. The observed increase in these molecules suggests that ESM treatment may promote cartilage repair by stimulating the synthesis of a new cartilage matrix. Our findings are consistent with previous studies demonstrating that eggshell membrane from hatched eggs increased the expression of collagen type II in H_2_O_2_-induced oxidative OA-like chondrocytes [48]. Additionally, studies with hydrolyzed type I collagen have shown its ability to promote collagen type II production in bovine chondrocytes [49]. These findings suggest that ESM and related compounds may share the ability to stimulate cartilage matrix synthesis.

This study provides promising evidence for the potential of ESM as a therapeutic agent for OA. However, some limitations need to be addressed in future studies. First, this study utilized an animal model, and further research is necessary to translate these findings to the human context. Clinical trials are warranted to confirm the efficacy and safety of ESM in human OA patients. Additionally, the optimal dose of ESM for long-term OA management needs to be determined. Second, while this study provides initial insights into the potential mechanisms of ESM’s action, further investigation is required to fully elucidate the detailed molecular pathways involved in its anti-inflammatory and cartilage-protective effects. This could involve exploring the effects of ESM on specific signaling pathways within chondrocytes, the cartilage-forming cells.

Overall, this revised version incorporates relevant previous studies on ESM and OA, strengthening the discussion and highlighting the potential of ESM as a therapeutic strategy.

## 5. Conclusions

In conclusion, this study demonstrates the potential of oral ESM administration as a therapeutic strategy for OA. Our findings in MIA-induced OA mice and rats showed that ESM treatment, particularly at high doses, alleviated pain and protected cartilage from degradation. The pain-relieving effect of ESM might be attributed to its ability to suppress the expression of pro-inflammatory cytokines and inflammatory mediators in the early stages of OA. Furthermore, ESM treatment downregulated the expression of matrix metalloproteinases (MMPs), enzymes responsible for cartilage breakdown, in the late stages of OA. Additionally, ESM stimulated the synthesis of new cartilage components (COL2α1 and ACAN) by increasing the expression of the transcription factor SOX-9. These findings suggest a multifaceted mode of action for ESM in OA, potentially involving both anti-inflammatory and cartilage-anabolic effects. While the precise mechanisms require further investigation, our study provides compelling evidence supporting the exploration of ESM as a potential therapeutic agent for OA.

## Figures and Tables

**Figure 1 nutrients-16-01885-f001:**
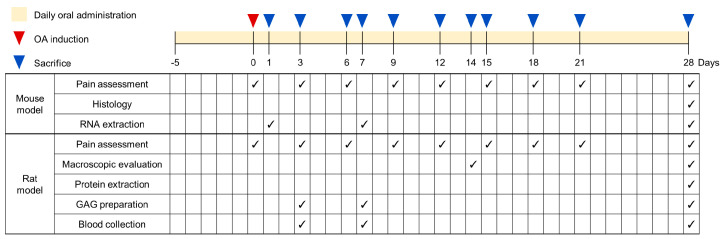
Schematic diagram of the experimental design and schedule for the mouse model and rat model. The animals (*n* = 30 per group) received a daily oral gavage of normal saline (vehicle), with increasing doses of ESM, or CLX for 5 days before OA induction (day 0), which continued 28 days post-induction. Pain assessments were performed every 3 days until day 21 in both the mouse and rat models, with a final assessment on day 28. Animals were then euthanized at the designated endpoints for knee joint tissue or blood sample collection. In the mouse model, knee joint articular cartilage tissues were collected from eight animals per group on days 1, 7, and 28 for mRNA expression analysis. At the endpoint (day 28), knee joints were collected from six animals per group for histological analysis. In the rat model, knee joint articular cartilage tissues and blood samples were collected for different analysis: six to seven animals per group on days 14 and 28 for macroscopic observation of the cartilage surface; six animals per group on days 3 and 5 for analysis of cartilage degradation markers in tissue and blood samples; and the remaining animals on day 28 (endpoint) for protein expression analysis and additional analysis of cartilage degradation markers. √ refers to the day of measurement.

**Figure 2 nutrients-16-01885-f002:**
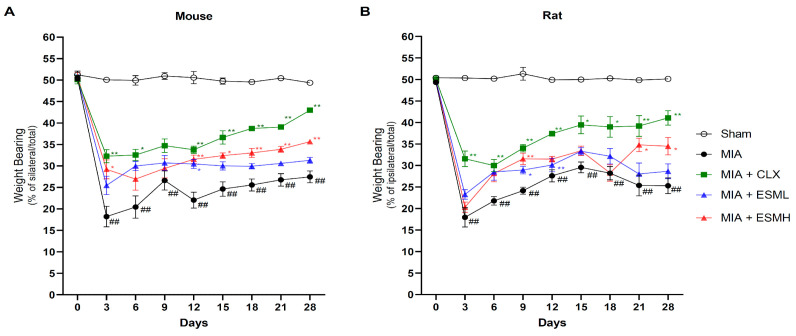
Effect of ESM on hind paw weight-bearing distribution in rodent models of MIA-induced osteoarthritis. Weight-bearing distribution in the hind paws of mice (*n* = 10 per group) (**A**) and rats (*n* = 8 per group) (**B**), as an indicator of joint pain, was measured every 3 days until day 21, with a final assessment on day 28. Values represent the percentage of weight borne by the ipsilateral paw relative to the total weight borne by both paws. Data are presented as mean ± SEM. Statistical significance was determined using repeated-measure (RM), two-way ANOVA followed by Dunnett’s post hoc test. Significant differences are indicated as follows: ^##^ *p* < 0.01 compared to Sham; * *p* < 0.05 compared to MIA; and ** *p* < 0.01 compared to MIA.

**Figure 3 nutrients-16-01885-f003:**
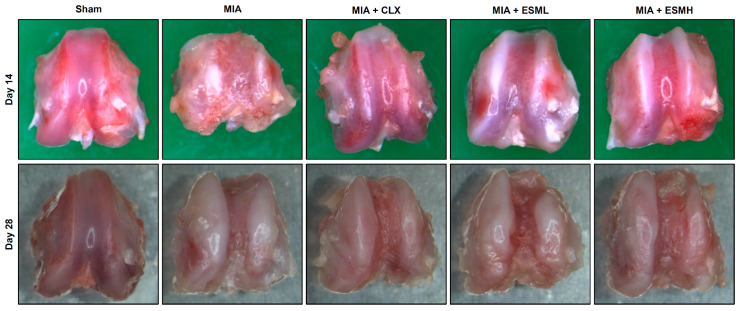
Effect of ESM on macroscopic changes of the articular cartilage surface in MIA-induced OA rats. Femoral cartilage was isolated from the knee joint of rats on days 14 and 28 post-OA induction and examined using a stereo-microscope (magnification: 40×). Images are representative of each experimental group (*n* = 6–7 per group).

**Figure 4 nutrients-16-01885-f004:**
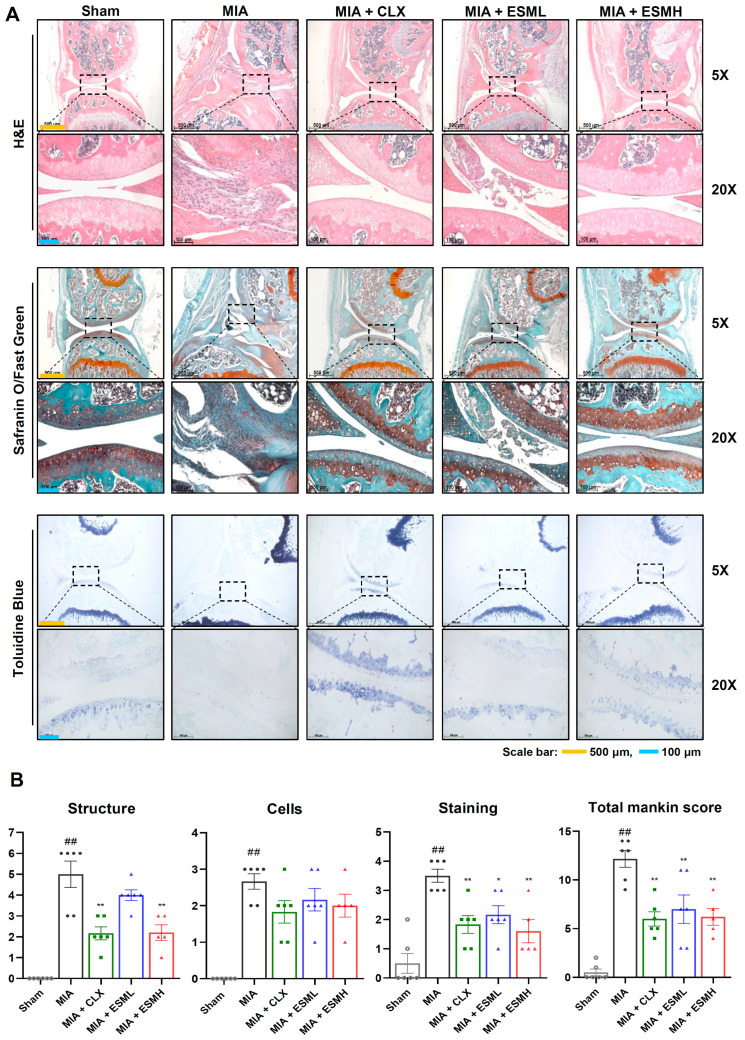
Histological evaluation of cartilage protective effect of ESM in MIA-induced OA mice. (**A**) Knee joint cartilage sections of the mice were stained with hematoxylin and eosin (H&E) (upper panels), Safranin O/Fast Green (middle panels), and Toluidine Blue (lower panels) to evaluate structural cartilage damage. The section images were captured under a light microscope at 5× and 20× magnification. Images are representative of each experimental group (*n* = 6 per group). (**B**) The lesions were graded using the modified Mankin scoring system, giving a combined total Mankin score. Data are presented as mean ± SEM (*n* = 6 per group). Statistical significance was determined using unpaired Student’s *t*-test. Significant differences are indicated as follows: ^##^ *p* < 0.01 compared to Sham; * *p* < 0.05 compared to MIA; and ** *p* < 0.01 compared to MIA.

**Figure 5 nutrients-16-01885-f005:**
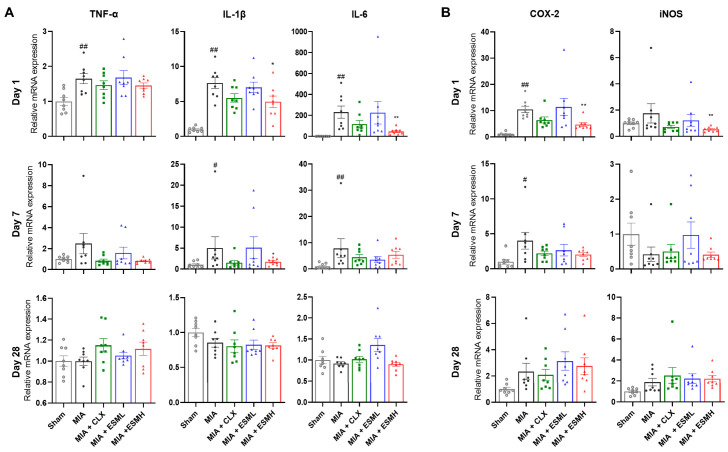
Effect of ESM oral administration on pro-inflammatory cytokines and inflammatory mediators in MIA-induced OA mice. Mice cartilage tissue samples were obtained on days 1, 7, and 28 post-OA induction. The mRNA expression of pro-inflammatory cytokines including TNF-α, IL-1β, and IL-6 (**A**) and inflammatory mediators such as COX-2 and iNOS (**B**) were determined using RT-qPCR. Data were normalized to ß-actin expression levels. Data values are presented as mean ± SEM (*n* = 6 per group). Statistical significance was determined using unpaired Student’s *t*-test. Significant differences are indicated as follows: ^#^ *p* < 0.01 compared to Sham; ^##^ *p* < 0.01 compared to Sham; * *p* < 0.05 compared to MIA; and ** *p* < 0.01 compared to MIA.

**Figure 6 nutrients-16-01885-f006:**
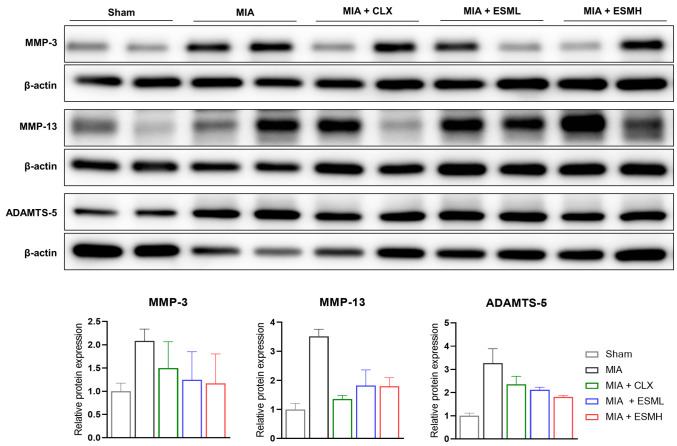
Effect of ESM oral administration on cartilage-degrading enzyme expression in the late stage of MIA-induced OA rats. Articular cartilage tissue samples were collected on day 28 post OA induction for protein expression analysis. Western blot analysis with densitometry quantification was used to measure protein levels of MMP-3, MMP-13, and ADAMTS-5. Protein loading was normalized to ß-actin. Representative Western blots from each group (*n* = 6–7 per group) are shown. Data are presented as mean ± SEM. Compete Western blots are presented in Appendix A.

**Figure 7 nutrients-16-01885-f007:**
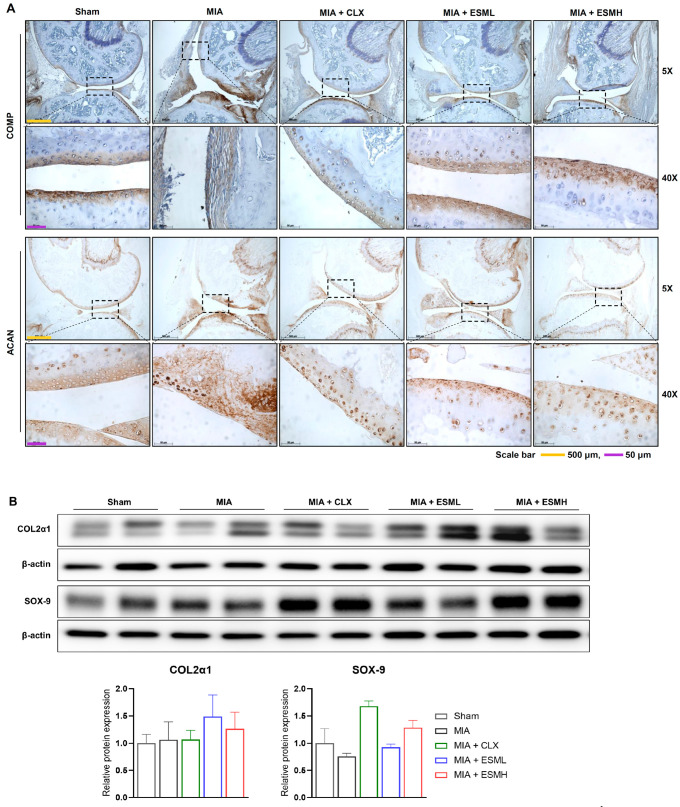
Effect of ESM oral administration on cartilage components and synthesis transcription factor in MIA-induced OA mice and rats. (**A**) Immunohistochemistry with a COMP antibody (upper panels) and ACAN antibody (lower panels) was performed on knee joint cartilage sections of the mice were to evaluate cartilage components. Section images were captured under a light microscope at 5× and 40× magnification. Images are representative within each experimental group (*n* = 6 per group). (**B**) Western blot analysis with densitometry quantification was used to measure protein levels of COL2α1 and SOX-9. Representative Western blots from each group (*n* = 6–7 per group) are shown. Data are presented as mean ± SEM. Compete Western blots are presented in Appendix A.

**Table 1 nutrients-16-01885-t001:** Treatment groups and doses.

Species	Groups	Substance	Dose (mg/kg/day)	Volume (mL)
Mouse	Sham	Saline Saline	-	0.1 0.1
	MIA		-	
	CLX	CLX	20	0.1
	ESML	ESM	60	0.1
	ESMH	ESM	120	0.1
Rat	Sham	Saline	-	1
	MIA	Saline	-	1
	CLX	CLX	10	1
	ESML	ESM	30	1
	ESMH	ESM	60	1

**Table 2 nutrients-16-01885-t002:** List and sequences of real-time PCR primers for mRNA expression.

Gene	Forward/Reverse	Sequence (5′→3′)
TNF-α	Forward	AGGGTCTGGGCCATAGAACT
	Reverse	CCACCACGCTCTTCTGTCTA
IL-1β	Forward	CTCGCAGCAGCACATCAACAAG
	Reverse	CCACGGGAAAGACACAGGTAGC
IL-6	Forward	ACAAAGCCAGAGTCCTTCAGAGAG
	Reverse	TTGGATGGTCTTGGTCCTTAGCC
COX-2	Forward	TGAGCAACTATTCCAAACCAGC
	Reverse	GCACGTAGTCTTCGATCACTATC
iNOS	Forward	GAGACAGGGAAGTCTGAAGCAC
	Reverse	CCAGCAGTAGTTGCTCCTCTTC
β-actin	Forward	AGCCATGTACGTAGCCATCC
	Reverse	CTCTCAGCTGTGGTGGTGAA

## Data Availability

Data are contained within the article.

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
