# Peer review of "Time-Dependent Effect of Eggshell Membrane on Monosodium-Iodoacetate-Induced Osteoarthritis: Early-Stage Inflammation Control and Late-Stage Cartilage Protection"

_nutrients, 2024, doi:10.3390/nu16121885_

Round 1

Reviewer 1 Report

Comments and Suggestions for Authors

The paper presents interesting research, which may have some practical, nutritional significance. The experiment was well designed and the manuscript was well written. There are some issues, which need to be corrected or explained:

1. line 50 - what is ECM? Please explain this abbreviation.

2. Introduction is definitely too long. There are to many biochemical and molceular details, which are widely known. Please reduce this part. 

3. lines 151-155 - this description refers to the production process performed by Eggnovo? 

4. The authors conducted research in an animal model. However, how might this research translate to humans? Is it possible to estimate how many eggs should be consumed to achieve the effects described in this work? Please refer to this in the discussion.

Author Response

Reviewer#1

The paper presents interesting research, which may have some practical, nutritional significance. The experiment was well-designed and the manuscript was well-written. There are some issues, which need to be corrected or explained:

Comment 1: Line 50 - what is ECM? Please explain this abbreviation.

Author’s Response: To avoid any confusion, we apologize that the abbreviation ECM (extracellular matrix) was used in the manuscript without prior definition. We have now corrected this oversight and included the definition of ECM at its first mention in the text.

Comment 2: Introduction is definitely too long. There are too many biochemical and molecular details, which are widely known. Please reduce this part. 

Author’s Response: We agree with the reviewer's suggestion to focus on the gap in knowledge relevant to our research topic. To achieve this, we have significantly reduced the Introduction section, focusing on key content directly related to our study.

Comment 3: Lines 151-155 - this description refers to the production process performed by Eggnovo?

Author’s Response: Yes. The ESM material was produced and provided by Eggnovo S.L., a leading European ESM producer and supplier. The same lot number of ESM was used throughout the entire study. We have revised the manuscript to clarify this point.

Comment 4: The authors conducted research in an animal model. However, how might this research translate to humans? Is it possible to estimate how many eggs should be consumed to achieve the effects described in this work? Please refer to this in the discussion.

Author’s Response: We appreciate the reviewer's comment. The ESM doses for each animal species were determined by following a practice guide for dose conversion between animals and humans (Nair, A.B.; Jacob, S. A simple practice guide for dose conversion between animals and human. Journal of basic and clinical pharmacy 2016, 7, 27). This guide recommends converting the recommended human equivalent daily doses of 300 and 600 mg. We have incorporated this information into the ‘2.3. Experimental Design and Administration’ section (lines 119-122) for clarity.

Regarding the reviewer's question about the number of whole eggs needed to achieve the effects of ESM, here's the answer: While approximately 20-30mg of dried ESM can be obtained from a single egg, it would require 10-20 eggs to reach the suggested daily doses for human. Importantly, our study utilizes purified and powdered ESM, not whole eggs. Whole eggs contain various components, including egg white and yolk, and are not necessary to achieve the effects observed in this study.

We appreciate the reviewer raising this important point for clarification.

Reviewer 2 Report

Comments and Suggestions for Authors

Dong et al. submitted the manuscript entitled: Eggshell Membrane Alleviating   Pain Behavior and Attenuates Disease Progression by Reducing Inflammatory Response in the Early Stage and Cartilage Degradation in the Late Stage on Monosodium Iodoacetate-Induced Osteoarthritis, in which they investigated on anti-osteoarthritis effects of ESM powder. The authors addressed on osteoarthritis phenotypes first, focusing on pain relieve and prevention of cartilage degradation. Further, the authors identified primary mechanisms, which mainly rely on down-regulation of pro-inflammatory factors and cartilage degradation-related enzymes. Generally, this manuscript was well prepared and this topic will be of interest to potential readers of Nutrients.

I have some minor comments for authors to improve the manuscript.

1. Figure 7B, for the significant level data in SOX-9, it’s a little unreasonable MIA was significant against Sham group while MIA+CLX group was not significant.  From current WB data, I tend to believe ESM showed neglectable effect on COL2α1 and SOX-9 expression.

2. The author mentioned both prevention of cartilage degradation and increase of cartilage synthesis. Did the increase of cartilage synthesis come from inhibition of pro-inflammatory factors or other mechanisms?

3. Please state the potential interest conflict.

4. Add full name of acronym ECM.

Author Response

Reviewer#2

Dong et al. submitted the manuscript entitled: Eggshell Membrane Alleviating   Pain Behavior and Attenuates Disease Progression by Reducing Inflammatory Response in the Early Stage and Cartilage Degradation in the Late Stage on Monosodium Iodoacetate-Induced Osteoarthritis, in which they investigated on anti-osteoarthritis effects of ESM powder. The authors addressed on osteoarthritis phenotypes first, focusing on pain relieve and prevention of cartilage degradation. Further, the authors identified primary mechanisms, which mainly rely on down-regulation of pro-inflammatory factors and cartilage degradation-related enzymes. Generally, this manuscript was well prepared and this topic will be of interest to potential readers of Nutrients.

I have some minor comments for authors to improve the manuscript.

Comment 1: Figure 7B, for the significant level data in SOX-9, it’s a little unreasonable MIA was significant against Sham group while MIA+CLX group was not significant.  From current WB data, I tend to believe ESM showed neglectable effect on COL2α1 and SOX-9 expression.

Author’s Response: Thank you for the suggestion regarding COL2α1 and SOX-9 expression. We have revised this section to emphasize the potential effects of ESM on these markers and included additional data in Supplementary Figure S3, which shows complete Western blots from individual animals in each experimental group. While our in vivo functional tests demonstrated promising results, the current data on COL2α1 and SOX-9 expression did not reach statistical significance. We acknowledge this limitation and will consider further investigation into these specific markers in future studies. We believe the overall findings, including the functional data and the observed trends in COL2α1 and SOX-9 expression, warrant further exploration of ESM's potential therapeutic effects in OA. We appreciate your insightful comments and hope you will find the revised manuscript suitable for publication.

Comment 2: The author mentioned both prevention of cartilage degradation and increase of cartilage synthesis. Did the increase of cartilage synthesis come from inhibition of pro-inflammatory factors or other mechanisms?

Author’s Response: Our study investigated the efficacy of ESM in alleviating pain and halting disease progression in MIA-induced rodent models of OA (mice and rats). We posit that the pain-relieving effect of ESM might be due to its ability to suppress the expression of pro-inflammatory cytokines and inflammatory mediators during the early stages of OA. Furthermore, ESM treatment downregulated the expression of matrix metalloproteinases (MMPs), enzymes responsible for cartilage breakdown, in the late stages of OA. Additionally, ESM stimulated the synthesis of new cartilage components (COL2α1, ACAN) by increasing the expression of the transcription factor SOX-9. These findings suggest a multi-faceted mode of action for ESM in OA, potentially involving both anti-inflammatory and cartilage-anabolic effects. We appreciate the reviewer raising this important point for clarification.

Comment 3: Please state the potential interest conflict.

Author’s Response: The manuscript was modified to address the reviewer’s comment.

Comment 4: Add full name of acronym ECM.

Author’s Response: To avoid any confusion, we apologize that the abbreviation ECM (extracellular matrix) was used in the manuscript without prior definition. We have now corrected this oversight and included the definition of ECM at its first mention in the text.
